

# Statistical Analysis on the Estimations of Solid Hydrometeors Growth Zones and Their Weather Conditions Using Radar Spectrum Width

**Sung-Ho Suh[1*], Woonseon Jung[2], Hong-Il Kim[1], Eun-Ho Choi[1], and Jung-Hoon Kim[3]**

Institutional addresses:

[1]Flight Safety Technology Division, NARO Space Center, Korea Aerospace Research Institute (KARI), 508 Haban-ro, Bongrae-myeon, Goheung-gun, Jeollanam-do, Republic of Korea

[2]Research Applications Department, National institute of meteorological sciences, 33, Seohobuk-ro, Seoqwipo-si, Jeju-do, 63568, Republic of Korea

[3]School of Earth and Environmental Sciences, Seoul National University, Seoul 08826, Republic of Korea

*Corresponding author: Dr. Sung-Ho Suh (suhsh@kari.re.kr)

**Keywords:** Spectrum width, Aerodynamic properties, Solid Hydrometeors, Weather radar, Dendritic Growth Zone (DGZ), Needle Growth Zone (NGZ), Growth Zone Determination Algorithm (GZDA).





**Abstract**
This study analyzes the correlation between hydrometeor type and radar spectrum width ($\sigma_v$)
according to wind speed that can occur the atmospheric disturbances such as turbulence and wind shear.
The $\sigma_v$ zones shown as peak values were identified only in stratiform precipitation and they are highly
related to the hydrometeor growth zones. Statistical analysis was performed for eight precipitation
cases under various conditions (precipitation type, season), focusing on the Dendrite Growth Zone
(DGZ) and the Needle Growth Zone (NGZ), where Dendrite (DN) and Needle (NE) type snowflakes
are dominant, respectively. They were determined by the Growth Zone Determination Algorithm
(GZDA) that was proposed in this study.
The intensity of the $\sigma_v$ depends on atmospheric conditions (i.e., wind speed) and season (i.e.,
temperature). The high $\sigma_v$ and negative relationship with the differential radar reflectivity ($Z_{DR}$) in the
DGZ for all cases is consistent with the aerodynamic properties of DN. As the range of $\sigma_v$ was larger
than that of $Z_{DR}$, it was confirmed that the dependence of $\sigma_v$ according to atmospheric conditions is
significant. Contrastingly, the NGZ had a low $\sigma_v$ and weak $\sigma_v$-$Z_{DR}$ negative relationship with a narrow
range of $\sigma_v$, which is consistent with the aerodynamic properties of NE. The lower cross-correlation
coefficient ($\rho_{hv}$) in the DGZ than that in the NGZ implies that the irregularities (particle shape and
aerodynamics features) of DN were more pronounced than those of NE.
Finally, as the altitudes of two growth zones were determined by temperature, the possibility of
estimating sub-zero temperature by GZDA was confirmed.



## 1. Introduction

Naturally formed ice crystal has various shapes. The International Commission on Snow and Ice describes seven ice crystal types: needles, columns, capped columns, plates, stellar crystals, spatial dendrites, and irregular forms (Mason, 1971). Libbrecht (2006) mentioned the field guide to the 35 type of snowflakes by 35 type with irregular snowflake. Hydrometeor identification helps in various fields such as I) remote sensing, in terms of quantitative precipitation estimation (e.g., Giangrande and Ryzhkov, 2008; Kennedy and Rutledge, 2011; Bechini et al., 2013), II) understanding the mechanisms of lightning formation (e.g., Ribaud et al., 2016), and III) aviation safety (e.g., Williams et al., 2011; 2013).

The solid hydrometeor formations that develop are determined by the water vapor pressure and atmospheric temperature (T). Ice crystals can develop with fine dust particles at $T > -40$ °C. The dendrites growth zone (DGZ) can be found at an altitude (H) where T range between $-20$ °C $< T < -10$°C, while the needles growth zone (NGZ) can be found at H where between $-5$ °C $< T < 0$ °C (Nakaya and Terada, 1935). Aircraft icing, which causes severe aviation problems, generally occurs at the altitude at which supercooled water droplets are present, corresponding to the range of $-20$ °C $< T < 0$ °C (Gent et al., 2000; Politovich et al., 2003). This indicates that the icing phenomenon may occur between two growth zones (GZs); thus, identifying GZs during flight is crucial for aviation safety.

The physical condition of the particle (i.e., size, shape, and density), as well as atmospheric condition (i.e., dynamic viscosity, atmospheric density), influences particle movements (i.e., vibration, orientation, and tumbling). Previous studies (e.g., Nakaya and Terada, 1935; Willmarth et al., 1964; List and Schemenauer, 1971; Ji and Wang, 1991; Wang and Ji, 1997; 2000; Wang, 2002; Hashino et al., 2014; 2016) have explained that particle behavior depends on shape, implying that a particle can exhibit various motions even under the same atmospheric condition. This further implies that radar velocity spectrum width ($\sigma_v$) may also depend on the shape and this assumption can be resolved if high



spatiotemporal measurements exist in a zone where each hydrometeor is homogenized (i.e., GZs).
Dual-polarization (dual-pol) weather radar supports high spatiotemporal measurements for analyzing
information on the sizes, shapes, and movements of various hydrometeors. Weather radar has an
excellent performance detecting and analyzing solid hydrometeors (e.g., Vivekanandan et al., 1994;
Ryzhkov et al., 1998; Wolde and Vali, 2001; Williams et al., 2011; 2013). Therefore, the compositions
in the DGZ (Kennedy and Rutledge, 2011; Andrić et al., 2013; Bechini et al., 2013; Suh et al., 2023)
can be explained by these radar products. Suh et al. (2023) demonstrated that the major radar products
with which to analyze GZs include their differential radar reflectivity ($Z_{DR}$ in dB), cross-correlation
coefficient ($\rho_{hv}$), and velocity spectral width ($\sigma_v$ in m s$^{-1}$). The differential radar reflectivity explains
the particle's oblateness (e.g., oblate, prolate). In contrast, the composition of hydrometeor types and
orientation of particles within the radar observation bin can be described by $\rho_{hv}$. The radar velocity
spectrum width, one of the major products obtained by Doppler weather radar, represents the deviation
in target movement within the observed resolution volume (Brewster and Zrnić, 1986; Doviak, 2006;
Zhang et al., 2009).
Researchers have attempted to find the relationship between turbulent motion and $\sigma_v$ (Labitt et al.,
1981; Knupp and Cotton, 1982; Hocking, 1985; Istok and Doviak, 1986; Jacoby-Koaly et al., 2002;
Melnikov and Doviak, 2009). Zhang et al. (2009) described the eddy dissipation rate (EDR) derived
by $\sigma_v$ from 14 cases with different weather conditions observed in the Hong Kong airport and evaluated
by aircraft measurements. Recently, Kim et al. (2021) suggested a new technique to estimate the $\sigma_v$-
based EDR using a lognormal mapping algorithm. The estimates show a high correlation with the
quick access recode data supported by the commercial aircraft and the EDR calculated by numerical
weather prediction. Suh et al. (2023) suggested that $\sigma_v$ can depend on the hydrometeor type based on
the features of radar products for GZs, which has been affirmed through simulations for dendrite (DN)
and needle (NE) types of snowflakes. However, they explained the need for further analysis as their



study I) considered only two winter precipitation cases and II) their GZs were extracted qualitatively.
As a follow-up to the study of Suh et al. (2023), this study suggests that representative features of $\sigma_v$
and dual-pol radar variables according to the atmospheric conditions [i.e., T, wind speed ($v$ in m s$^{-1}$)]
in GZs improve the practicality of the present results based on statistical analysis using eight
precipitation cases with various conditions (i.e., season, precipitation type), and a quantitative
approach through a Growth Zone Determination Algorithm (GZDA). In addition, the possibility of
estimating T above freezing levels with weather radar was examined by the GZDA. The remainder of
this paper is presented as follows: Section 2 provides details on the data and instruments, including the
structure of the GZDA. The results of the radar products extracted by the GZDA and their performance
are investigated in Section 3, with a discussion of said results found in Section 4, while the conclusions
of this study are presented in Section 5.



## 2. Data and Methods

### 2.1. Analysis instrument and data

The Yongin Testbed dual-pol weather radar (YIT) was used to analyze the features of GZs (Fig. 1). It has been operated by the Weather Radar Center, under the Korean Meteorological Administration (KMA), from July 2014 until the present. The Enterprise Electronics Corporation manufactured the YIT with coverage of up to 240 km and an S-band (2.88 GHz) transmission frequency. Specifications regarding the YIT are presented in Table 1.

The cases analyzed in Suh et al. (2023) were limited to winter stratiform precipitation, while the scope of the present study was expanded to examine the radar products in GZs for eight precipitation cases which can be divided into I) winter stratiform precipitation, and II) non-winter precipitation. Non-winter precipitation cases were subdivided into stratiform and convective types (Table 2). They are expressed as abbreviations to describe the characteristics of cases with various conditions intuitively. The first letter of the abbreviation means the precipitation type [Stratiform (S) and Convective (C)], and the second letter means the season [Spring (P), Summer (S), and Winter (W)]. The last digit is the number indicating the case. From the PPI for the lowest elevation angle ($\theta$), if there is a significant cell that satisfies the radar reflectivity ($Z_H$) > 40 dBZ within 1km of vertical altitude from the ground or that satisfies $Z_H$ > 45 dBZ within 250 km radar slant range, it was considered as convective precipitation. The research cases selected satisfied the following conditions: I) the precipitation occurred for at minimum over three hours at the YIT site, and II) the vertical depth of $Z_H$ in a precipitation system higher than 5 dBZ exceeded 2 km (Fig. 2).

Meteorological information was obtained from the mesoscale model (MSM) reanalysis data to analyze the radar products according to atmospheric conditions in GZs. The MSM provided by the Japanese Meteorological Agency supports meteorological parameters for sixteen pressure altitudes. As



this reanalysis data has a better spatial resolution of pressure altitudes increases as it is closer to the
ground, it is useful to interpret the atmospheric information for altitudes below the middle level (500
hPa).

**2.2. Analysis strategies**
**2.2.1. QVP**
The quasi-vertical profile (QVP) technique proposed by Ryzhkov et al. (2016) was applied to define
and analyze the GZs. The QVP is a computationally efficient scheme because it can be calculated with
only one sweep. Radar products in QVP are obtained by azimuth averaging and are expressed in a
time-height format. Moreover, this technique helps analyze the features of hydrometeors (Ryzhkov et
al., 2016). The slant ranges at the three target elevation angles (19°, 17.3°, and 16.4°) for the altitude
of 12 km without consideration of the curvature of the earth are 36.85 km, 40.12 km, and 42.50 km,
respectively. A slant range of approximately 2.5 km from YIT is considered a dead zone, corresponding
to H of approximately 0.8 km at $\theta = 19°$, the highest elevation level in the selected cases.

**2.2.2. GZ Determination Algorithm (GZDA)**
This study determined the GZ for several precipitation cases, including various seasons, through
quantitative criteria in a deductive approach to analyze the statistical characteristics of the GZs. A total
of 5 stages were conceived in the GZ Determination Algorithm (GZDA) to identify GZs for all
precipitation cases (Fig 3).
STAGE 1 (Pre-processing): Firstly, the QVP was created and then the median value of $\sigma_v$ ($\sigma$) and its
variation ($d\sigma$) for each vertical window channel in the whole analysis periods were calculated. After



that σ and dσ are smoothed for ±1 vertical window channel to define the peak of dσ (κ) and its altitude
($H_\kappa$).
STAGE 2 (κ and $H_\kappa$ determination): To remove the noise in κs, only one κ which corresponds to the
extreme value in ±2 vertical window channel was selected.
STAGE 3 (Freezing level considerations): The region of interest in this study was an altitude at
which the temperature is below freezing. κ can be found not only on the freezing layer but also on the
melting layer (ML), negatively affecting the correct determination of σ and dσ. Thus, κ was identified
and removed from the bottom of the ML to the ground before procedures were implemented on the
freezing level, which was the target of analysis. This procedure for the unfreezing level should be
prioritized since a minimum κ may be found at the bottom of the ML rather than near the bottom of
the NGZ.
STAGE 4 (GZ determination): κ that satisfied the characteristics of interest shown in GZs were
selected and defined as their boundary. To accomplish this, extreme values of σ that can be considered
as the core of each GZ within the freezing level were chosen. Then, the altitudes where the minimum
and maximum values of σ ($H_{\sigma 1}$ and $H_{\sigma 2}$, where $H_{\sigma 1} < H_{\sigma 2}$) exist, representing the cores of the NGZ and
the DGZ, respectively, were found. This means that the peak values of κ are matched to the boundary
altitudes of GZs. The altitudes at the bottom and top of the DGZ (the NGZ) have the local maxima
(minima) and the local minima (maxima) of κ, respectively. After the two procedures described above
were complete, the four nearest $H_\kappa$s ($H_{N1}$, $H_{N2}$, $H_{D1}$, and $H_{D2}$) from the peak value of σ could be
designated as the boundary list. These four points have to be satisfied the following conditions: I) $H_{N1}$
$\leq H_{\sigma 1} \leq H_{N2}$, and II) $H_{D1} \leq H_{\sigma 2} \leq H_{D2}$.
STAGE 5 (Post-process): The final step was to determine whether the selected GZs met realistic
conditions and to supplement the results that were not resolved in Stage 4. When only the bottom of



the DGZ ($H_{D1}$) was chosen as the boundary list for DGZ due to the limitations of the radar observation
strategy, the echo top height ($H_T$) was chosen as the top of the DGZ ($H_{D2}$). In addition, since it is rare
for the thickness of GZ to exceed 1.5 km in natural conditions, the GZs that satisfies this condition
may unreliable, so it was excluded. A detailed description of the algorithm with the overall procedure
was summarized in Figure 5.

**2.3. Data quality control**
Initially, a fuzzy logic algorithm was applied (Gourley et al., 2007) to radar products to remove non-
meteorological targets. The $\sigma_v$ was removed when it satisfied a signal-to-noise ratio (SNR) condition
where SNR < 20 dB because $\sigma_v$ shows a large variance for lower SNRs (Zhang et al., 2009). The
following three additional QC procedures for QVP data were performed after the aforementioned QC
procedures: I) the radar product in PPI was removed if the number of data points where $Z_H > 5$ dBZ
was less than 20 % of the total number of azimuths at the $i^{th}$ radar bin, and II) the QVP for $Z_H \leq 5$ dBZ
was selected, III) the calibrated $Z_{DR}$ was applied because the projected area of the target depends on
the line of sight. The $Z_{DR}$ calibration suggested by Ryzhkov et al. (2005) was introduced as follows,

$$Z_{DR}(\theta) = \frac{Z_{DR}(0)}{[Z_{DR}(0)^{1/2}sin^2\theta + cos^2\theta]^2}$$


The radar elevation angles for QVP considered in this study are $\theta = 16.4°$, $17.3°$, and $19.0°$, and the
calibrated $Z_{DR}$ for these $\theta$ are $0.916Z_{DR}$, $0.906Z_{DR}$, and $0.887Z_{DR}$ for $\theta=0°$, respectively.





## 3. Results

### 3.1. Analysis of QVPs

#### 3.1.1 QVP for various conditions of precipitation cases

**3. Results**
**3.1. Analysis of QVPs**
**3.1.1 QVP for various conditions of precipitation cases**
The features of QVP-based dual-pol radar variables are classified into the type of precipitation
(stratiform and convective precipitation). Seasonal factors can also subdivide the profile pattern for
stratiform precipitation. Figure 4 presents a representative case to explain the common feature of dual-
pol radar products for the non-winter stratiform precipitation case selected in this study. The following
common features can be identified from the QVP for the non-winter stratiform precipitation.
First, the altitude of melting layer can be confirmed at $H > 1$ km where the weather radar can clearly
detect height. High $Z_H$, $Z_{DR}$, and low $\rho_{hv}$ can be found in the ML due to the melting of solid
hydrometeors (Trömel et al., 2017). In addition, weak $Z_H$ and $\rho_{hv}$ appear in the upper layer of the ML,
while high $Z_H$ and $\rho_{hv}$ appear in the lower layer of the ML due to the increase in the dielectric constant
and the uniformity of particle shape. A low $Z_{DR}$ appears near the top of the ML and increases with H.
Further, in the layer of a degree above zero, a higher $Z_{DR}$ than near the top of the ML can be found,
which corresponds to a negative relationship between $Z_H$-$Z_{DR}$ (i.e., solid hydrometeor), indicating a
spherical hydrometeor (i.e., raindrop) with high $Z_H$ (i.e., high dielectric constant). There are no
remarkable features in $\sigma_v$ except for a weak increase in the ML.
By contrast, stratiform precipitation in winter can have only solid hydrometeors in the whole system
since strong vertical convection, which can transport supercooled water droplets to the upper layers,
have not developed (Fig. 5). Stratiform precipitation in winter has comparable characteristics to those
of the non-winter season. Firstly, there are no ML; thus, the entire precipitation system matches the
features in the upper layer of ML. Secondly, two $\sigma_v$ zones in the freezing level can be found, which are
distinctly developed on H where temperatures are -5 ℃ and -15 ℃, respectively.




### 3.1.2 GZ determination in QVP

The vertical profiles of $\sigma_v$ to which GZDA was applied for various precipitation cases are shown in
Figure 6. All GZs were determined correctly for the stratiform precipitation group, and there were no
GZs determined by GZDA for precipitation cases classified as a convective type.
In the vertical profile of $\sigma_v$ for stratiform precipitation in winter, extreme values of $\sigma$ can be
confirmed, which were statistically significant for cases of SW3 and SW4. Minimum and maximum
values of $\sigma$ are observed for NGZ and DGZ, respectively, in all precipitation cases presented from
GZDA. In the case of SW1-4, it was found that the top of the DGZ, which was not identified due to
the characteristics of the case formed at a low altitude, clearly appeared in SP7. Through this, it can be
confirmed in the case of SP7 that the $\sigma_v$ in the freezing level is not proportional to the altitude. The
maxima of $\sigma$ is found around $3 < H$ (km) $< 4$ and $H = 4.5$ km for cases in the spring and summer
seasons, respectively, but it matches the altitude of the ML. Moreover, the vertical profiles of $\sigma_v$ in
these cases had a vertically symmetrical pattern centered on the ML. In addition, there were $\kappa$ in
convective precipitation cases determined by GZDA, but GZs are not presented since the $\kappa$ did not
satisfy the conditions of the algorithm. There are minima of $d\sigma_v$ in the GZ transition region for
stratiform precipitation cases but it not shown in figure since it is removed by the algorithm (Fig. 7).
Still, no minima of $d\sigma_v$ was observed in the region between 2[nd] and 3[rd] of the boundary list that can be
expected as a transition region of GZ for convective precipitation cases (CP6 and CS8) even though
they have four $\kappa$s.

### 3.2. Features of dual-pol variables in GZs



### 3.2.1 Dominant features in GZs


The characteristics of the dual-pol weather radar variables in the $\sigma_v$ zones defined from the GZDA
were different from those obtained in the altitudes where T = -15 °C and -5 °C, which corresponds to
the potential area of DGZ and NGZ for the stratiform precipitation case. The weather information
provided from the MSM reanalysis data was used. To select the possible area of GZs for convective
precipitation cases that were not defined by the GZDA, ranges of ±0.3 km (DGZ) and ±0.1 km (NGZ)
from the H where T = -5 °C and -15 °C were considered. The potential range of the GZs were defined
by the average range of the GZs identified in this study, but that of the NGZ were strictly defined
relatively narrow in order not to include the ML.
For the DGZ, $Z_{DR}$ for stratiform precipitation in winter was widely distributed around the positive
values (Fig. 8). Contrastingly, stratiform precipitation in the spring, and especially convective
precipitation, had a concentrated distribution with a narrow modal range of $0.4 < Z_{DR}$ (dB) $< 0.7$. The
cases of SW1 and SW2 showed a high $Z_{DR}$, with a modal $Z_{DR}$ of 1 dB due to a strong $Z_{DR}$ column. By
contrast, the cases of SW3 and SW4 had the lowest $Z_{DR}$ distribution among all cases, with modal
values ranging from $0.2 < Z_{DR}$ (dB) $< 0.3$. In terms of $\rho_{hv}$, stratiform precipitation in winter for cases
other than SW2 showed a wide and gentle distribution with a mode of $0.976 < \rho_{hv} < 0.986$. Contrarily,
those in non-winter precipitation had a relatively high mode $(0.983 < \rho_{hv} < 0.990)$ with a concentrated
distribution. The distribution of $\sigma_v$ had a negative relationship with that of $Z_{DR}$, which is consistent
with Suh et al. (2023). SW3 and SW4, which had a lower $Z_{DR}$, had the highest modal value, at about
$\sigma_v = 1.2$ m s$^{-1}$, while SW1 and SW2, which had a higher $Z_{DR}$, showed relatively low modes at about
$0.5 < \sigma_v$ (m s$^{-1}$) $< 0.8$. The distribution of $\sigma_v$ for non-winter precipitation cases also showed a negative
relationship with $Z_{DR}$, similar to the winter cases.
The $Z_{DR}$ in the NGZ showed a similar pattern to that of the DGZ (Fig. 9). The mode of $Z_{DR}$ in SW3





and SW4 was negative, while all other cases had positive values. In contrast, $\rho_{hv}$ and $\sigma_v$ in the NGZ
showed noticeably different distributions from those in the DGZ. First, the distribution of $\rho_{hv}$ was
concentrated with a mode of $0.930 < \rho_{hv} < 0.996$ for all cases except for SW3 and SW4, which showed
gentle distributions centered on $\rho_{hv} = 0.985$. In addition, it is characterized by having the highest $\rho_{hv}$
among all cases centered on $\rho_{hv} = 0.995$ in non-winter precipitation. These tendencies are also
confirmed in $\sigma_v$. The case of SW3 and SW4 had gentle distributions of $\sigma_v$, with a mode of $0.70$ m s$^{-1}$,
while all other cases showed a concentrated distribution with various modes ranging from $\sigma_v = 0.4$ m
s$^{-1}$ to $0.64$ m s$^{-1}$.

**3.2.2 Relationships of dual-pol variables**
Correlation analysis was performed to interpret the relationship between the dual-pol radar variables
and $\sigma_v$ which showed a dependence on precipitation type for each case in GZs (Fig. 10). $Z_{DR}$-$\sigma_v$ in the
DGZ showed a negative relationship for all cases (Fig. 10a). A high $\sigma_v$s of $1.2$ m s$^{-1}$ in SW3 and SW4
were found, where $\sigma_v$ gradually decreased as $Z_{DR}$ increased in other cases. $Z_{DR}$-$\sigma_v$ had various
relationships for each case, but generally, their distribution did not correlate. SW2 and SW3 had a
positive correlation for $Z_{DR}$-$\sigma_v$, but in the case of SW2, correct linear regression was not performed
since most of the data points were concentrated in $Z_{DR} \sim 1$ dB. In the case of SW3, the distribution of
$Z_{DR}$-$\sigma_v$ did not correlate. $\rho_{hv}$ did not show a significant correlation with $\sigma_v$, but $Z_{DR}$ showed a common
trend of inverse proportion to $\rho_{hv}$ in all cases.
The averaged $Z_{DR}$- $\sigma_v$ of each case have a negative relationship, and the variation range of $\sigma_v$ is
higher than that of $Z_{DR}$ (Fig. 10b). Additionally, $\rho_{hv}$ showed a negative relationship with $\sigma_v$.
Remarkably, the cases SW3 and SW4 displayed the lowest $\rho_{hv}$ and the highest $\sigma_v$. $Z_{DR}$- $\sigma_v$ had a linear
distribution and showed a remarkable correlation, with a root mean square error (RMSE) = $0.05$ m s$^{-}$

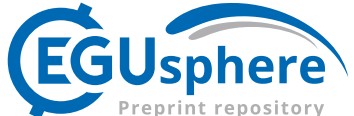



[1], in stratiform precipitation in winter. On the other hand, in all stratiform precipitation cases, $Z_{DR}$-$\sigma_v$
had a relatively low correlation of RMSE = 0.11 m s$^{-1}$. They had a significantly different distribution
in convective precipitation from stratiform precipitation in winter. Average $Z_{DR}$-$\sigma_v$ values for SP7 were
located in between stratiform precipitation in winter and convective precipitation.
In the case of the NGZ, $Z_{DR}$-$\sigma_v$ had a relatively weak negative correlation for all cases (Fig. 10c).
The variation range in $Z_{DR}$ was broader than that observed in the DGZ, ranging from -0.5 dB – 3 dB.
Also, $\rho_{hv}$ showed a negative correlation with both $Z_{DR}$ and $\sigma_v$. There was a negative $Z_{DR}$, while $\rho_{hv}$
increased overall ($\rho_{hv} > 0.98$) in the NDZ, compared to the DGZ. In contrast, $Z_{DR}$-$\sigma_v$ for each case had
a positive relationship except for convective precipitation, and their slope tended to depend on the $Z_{DR}$
for each case. The positive correlation between $Z_{DR}$ and $\sigma_v$ tended to increase as the $Z_{DR}$ for each case
decreased.
The averaged $Z_{DR}$-$\sigma_v$ of each case, for all cases in the NGZ had a relatively weaker negative
relationship than that in the DGZ (Fig. 10d). There was a remarkable variance in $Z_{DR}$ compared to that
of $\sigma_v$, differing from what is observed in the DGZ. The high $\sigma_v$ found in SW3 and SW4 also displayed
a considerable variation in $\sigma_v$, while SW1 and SW2 had a low $\sigma_v$ and a large variation in $Z_{DR}$. The
pattern in $\rho_{hv}$ and $\sigma_v$ corresponded to that of the DGZ, while the overall averaged $\rho_{hv}$ was higher than
that of the DGZ. A linear distribution with a striking correlation of RMSE = 0.03 m s$^{-1}$ was shown for
both stratiform precipitation in winter and total stratiform precipitation cases. In addition, a significant
difference from the DGZ is that the averaged $\sigma_v$-$Z_{DR}$ for convective precipitation is also located quite
close to the regression lines for those of stratiform precipitation.
The maturity of GZs is related to the ratio of dominant solid hydrometeor within each GZ. It appears
to have an inverse relationship with $\rho_{hv}$, and the lower $\rho_{hv}$ in the DGZ compared to NGZ was found
(Suh et al., 2023). The maturity of GZs can be indirectly explained by the difference in aerodynamic





characteristics (i.e., $\sigma_v$) between the core of GZ and that of boundaries ($d\sigma_{MAX}$). Overall, $d\sigma_{MAX}$ showed
a pattern inversely proportional to $\rho_{hv}$ (Fig. 11). There is a negative correlation between $d\sigma_{MAX}$ and $\sigma_v$.
For the range of $\sigma_v > 0.7$ m s$^{-1}$, only stratiform precipitation is confirmed, and conversely, for the range
of $\sigma_v < 0.7$ m s$^{-1}$ where convective precipitation dominates, the condition of $d\sigma_{MAX} < 0.03$ was
confirmed. The negative relationship between $d\sigma_{MAX}$ and $\rho_{hv}$ was more apparent in the NGZ. The NE
had the highest $d\sigma_{MAX}$ (0.13 and 0.21) among all cases in SW3 and SW4, where the averaged $Z_{DR}$ was
negative value. $d\sigma_{MAX}$s for stratiform precipitation in all cases except for SW3 and SW4 was found to
be between 0.08 and 0.11, while convective precipitation had a low $d\sigma_{MAX}$ of less than 0.04.

**3.3. Characteristics of dual-pol radar variables in GZs with atmospheric condition**
The dual-pol radar variables are determined by the maturity of GZs. In addition, atmospheric
disturbance affects the orientation of the hydrometeor, which increases $\sigma_v$ and decreases $\rho_{hv,}$ thus
influencing $Z_{DR}$. Consequently, $d\sigma_{MAX}$ can depend on aerodynamic conditions. It was confirmed by
the relationship between $v$ and $\sigma_v$ in GZs.
The $\sigma_v$ has a positive relationship with $v$ for all cases in the DGZ (Fig. 12). In particular, it displays
a marked correlation with an RMSE = 0.14 m s$^{-1}$ for only stratiform precipitation in winter. SP7 showed
a relatively weak $\sigma_v$ of 0.76 m s$^{-1}$ despite having a condition of strong $v$ of about 26 m s$^{-1}$. Accordingly,
the linear regression for all stratiform precipitation cases was less correlated (RMSE = 0.25 m s$^{-1}$) than
for stratiform precipitation in winter (RMSE = 0.14 m s$^{-1}$). In addition, convective precipitation showed
a relatively weak $\sigma_v$ of $0.48 < \sigma_v$ (m s$^{-1}$) $< 0.66$ despite a relatively strong $v$ of more than 17 m s$^{-1}$. SP7
was not included in the $\sigma_v$-$v$ relationship for stratiform precipitations because of the lower $\sigma_v$ even
though the strongest $v$ (26 m s$^{-1}$) among the whole case. This supports the explanation that seasonal
conditions influence the developmental characteristics of GZs. In addition, CS8 showed the lowest





correlation as it showed the lowest $\sigma_v$ (0.5 m s$^{-1}$) within the same $v$ range. NGZ was confirmed to be
uncorrelated to $\sigma_v$-$v$, as suggested by Suh et al. (2023). Interpretable correlations were not identified
for stratiform and convective precipitation since averaged $\sigma_v$s for all cases except for SW3, and SW4
can be found in $0.43 < \sigma_v$ (m s$^{-1}$) $< 0.67$ regardless of $v$.

**4. Discussion**
The growth rate of each hydrometeor type depends on the condition of T and Water vapor pressure
(Nakaya and Terada, 1935). Water vapor pressure depends on atmospheric pressure, meaning that the
development of GZs can be determined by altitude. That's why the intensity of $\sigma_v$ zone shown in the
present study appear to be inverse proportional to the altitude of GZ. The intensity of the $\sigma_v$ zone in
seasonal stratiform precipitation clearly verified this. Although there was more substantial average $v$
(26 m s$^{-1}$) compared to higher that of winter stratiform precipitation, SP7 showed relatively low $\sigma_v$
($\sim$0.76 m s$^{-1}$) because the DGZ for the stratiform precipitation in spring had a higher altitude (5.4 < H
(km) < 5.9) than that of the winter season (2.6 < H (km) < 4.1). Nevertheless, the atmospheric
temperature where each GZ was found remains consistent regardless of the intensity of $\sigma_v$ and its
altitude. It implies that as GZ altitude depends on atmospheric temperature, it suggests that radar-based
sub-zero T estimation from GZDA might be possible. The GZDA-based statistical analysis presented
here confirmed that stratiform precipitations in winter satisfied the temperature conditions, with -10 <
T (℃) < -16 and 0 < T (℃) < -5 in both the DGZ and the NGZ, respectively (Fig. 13). Once GZs have
been determined, a transition area between the GZs can be identified. This implies that the real-time
estimation of T for every 5 ℃ class interval from 0 ℃ to -15 ℃ by the weather radar will is possible.
It is expected that GZDA can be used for the purpose of flight safety since the layer where GZ can
develop is matched with the layer at which aircraft icing can occur.



As explained in the introduction section, EDR can be estimated from $\sigma_v$, representing the variation
of particle motion within the observation volume (e.g., Zhang et al., 2009; Kim et al., 2021). Suh et al.
(2023) suggested the possibility that $\sigma_v$ depended on hydrometeor types, which was confirmed in this
study. This means that $\sigma_v$ (EDR) can be varied by the hydrometeor types under the same atmospheric
conditions. As a result, the following two improvements are suggested, I) EDR correction: First, the
correction of $\sigma_v$-based EDR would be required if it is assumed that the motion of the hydrometeor
cannot represent atmospheric disturbance. That is, when estimating $\sigma_v$-based EDR, the process of
applying the EDR correction for each hydrometeor type has to be considered. II) EDR prediction: If it
is assumed that hydrometeor motion can represent atmospheric disturbance regardless of hydrometeor
type, then this implies that a prediction for radar-based EDR by $\sigma_v$ and hydrometeor classification
algorithm would be possible. Based on the strong correlation between the dendrite type and $\sigma_v$
identified in this study, it is expected that the area where strong atmospheric turbulence will occur can
be predicted. This will be especially helpful to improve flight safety as the potential area of high EDR
can be predicted.

**5. Summary and Conclusion**
This study was conducted to reduce the adverse effects that can be caused by meteorological
phenomena caused by disturbances such as shear/turbulence/icing in aircraft operations. Accordingly,
it is intended to help provide real-time weather information for safe navigation by utilizing weather
radar products. This has been verified from eight precipitation cases with different conditions
(precipitation type and season), using the dominant atmospheric characteristics of each hydrometeor
type that can be estimated from $\sigma_v$ to suggest the possibility of the GZ determination. The results from
a previous study on this topic (Suh et al., 2023) were verified in this study, and the key results of each



GZ with radar dual-pol variables in stratiform precipitation are as follows (Fig. 14).
Firstly, the variation range of $Z_{DR}$ in the DGZ was narrower than that of $\sigma_v$. This suggests that
although there are significant variations in $Z_{DR}$ due to particle orientation caused by an atmospheric
disturbance, the variation of $\sigma_v$ is more significant. A strong negative relationship between $\sigma_v$-$Z_{DR}$ in
all DGZ instances allows us to confirm the features in DN where both $Z_{DR}$ and $\sigma_v$ are strongly
influenced. Therefore, $\sigma_v$ increases as the turbulence becomes stronger (as $Z_{DR}$ decreases) for oblate
particles, but the dependence of $\sigma_v$ according to atmospheric conditions is more prominent than that of
$Z_{DR}$. The irregular particle shape such as DN can explain the negative relationship of $Z_{DR}$-$\rho_{hv}$.
Moreover, the DGZ has lower $\rho_{hv}$ compared to the NGZ. This indicates that the DN has unstable
movement due to aerodynamic features rather than NE due to an irregular shape.
A weak negative relationship of $\sigma_v$-$Z_{DR}$ in the NGZ for the all case can be inferred that this is due to
the combination of hydrometeors with NE as the major, which negligible influences on $\sigma_v$.
Theoretically, the variation of $\sigma_v$ in NE is negligible regardless of $v$ (e.g., Suh et al., 2023). Therefore,
the similarity of particle shape ($\rho_{hv}$) is higher in NGZ and means that aerodynamic properties in NE
are relatively more coherent. In cases of negative $Z_{DR}$ where NE might be sufficiently grown, the
inverse relationship of $\sigma_v$-$\rho_{hv}$ is enhanced. However, the $Z_{DR}$ for each observation case has a positive
relationship with $\sigma_v$ and a negative relationship with that of $\rho_{hv}$ in the NGZ. This implies that the NGZ
could have various hydrometeors that came from the upper layer, and these hydrometeors can be
formed by their interactions, secondary ice production, (e.g., Field et al., 2017).

## Author contributions


Dr. Sung–Ho Suh designed the study. Dr. Sung-Ho Suh and Dr. Woonseon Jung collected the
samples and performed the study. Dr. Sung–Ho Suh, Dr. Hong-Il Kim, and Eun-Ho Choi obtained the



results and prepared the manuscript with contributions from all the coauthors. Dr. Jung-Hoon Kim
examined the results and checked the manuscript. All authors have read and agreed to the published
version of the manuscript.

## 398 Competing interest

None.

## 401 Code/Data availability

The data obtained by YIT in this study are available on request from Korea Meteorological
Administration (KMA) and the codes are available on request from Dr. Sung-Ho Suh.

## 405 Acknowledgments

This research was supported by the Space Center Development Project (II) of Ministry of Science
and ICT (MSIT)




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



## Tables

**Table 1.** Specifications of Yongin Testbed (YIT).

| Specifications | Details |
| --- | --- |
| Model | DWSR-8501 S/K-SDP |
| Manufacturer | EEC (US) |
| Transmitting tube | Klystron |
| Antenna diameter | 8.5 m |
| Transmission frequency | 2.88 GHz |
| Peak power | 850 KW |
| Effective observation range | 240 km |
| Beam / Pulse width | 0.94° / 2 μs |
| Wavelength | 10.41 cm |
| Range gate size | 250 m |
| Elev. height | 473 m |
| Long. / Lat. | 127.2852 °E / 37.2063 °N |
| Obs. interval | 10 min |



**Table 2.** Information of precipitation cases selected in this study.

| No | Case | Date | Time (LST) | θ for QVP | Type | Season |
|----|------|------|------------|-----------|------|--------|
| 1 | SW1 | 16. Feb. 2015 | 0000-0600 | 16.4 | Stratiform | Winter |
| 2 | SW2 | 21. Feb. 2015 | 1000-1700 | 16.4 | Stratiform | Winter |
| 3 | SW3 | 27. Feb. 2016 | 0000-0600 | 19.0 | Stratiform | Winter |
| 4 | SW4 | 28. Feb. 2016 | 1200-1900 | 19.0 | Stratiform | Winter |
| 5 | CP5 | 02. May. 2016 | 1800-2400 | 17.3 | Convective | Spring |
| 6 | CP6 | 10. May. 2016 | 0600-1500 | 17.3 | Convective | Spring |
| 7 | SP7 | 24. May. 2016 | 0330-1100 | 17.3 | Stratiform | Spring |
| 8 | CS8 | 01. Jul. 2016 | 1330-1630 | 17.3 | Convective | Summer |








**Figures**

Figure 1. Origin of Yongin Testbed (YIT) dual-pol weather radar data (red bullet) in South Korea.




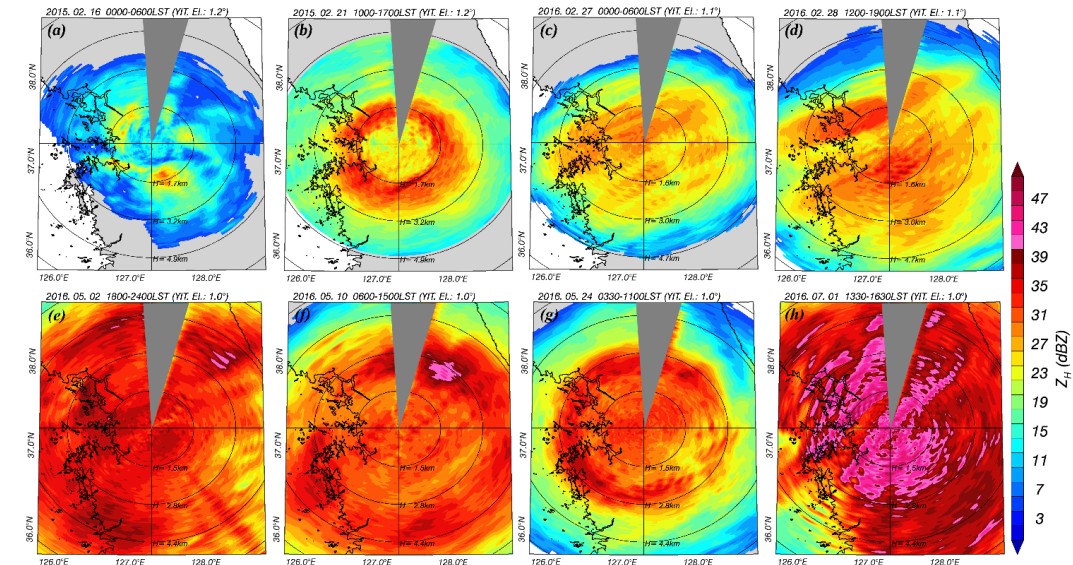


Figure 2. Cumulate maximum $Z_H$ in PPI at the lowest elevation angle for analysis cases. The grey

blank indicates a beam blockage area. The range rings are centered on the YIT radar at 50 km.




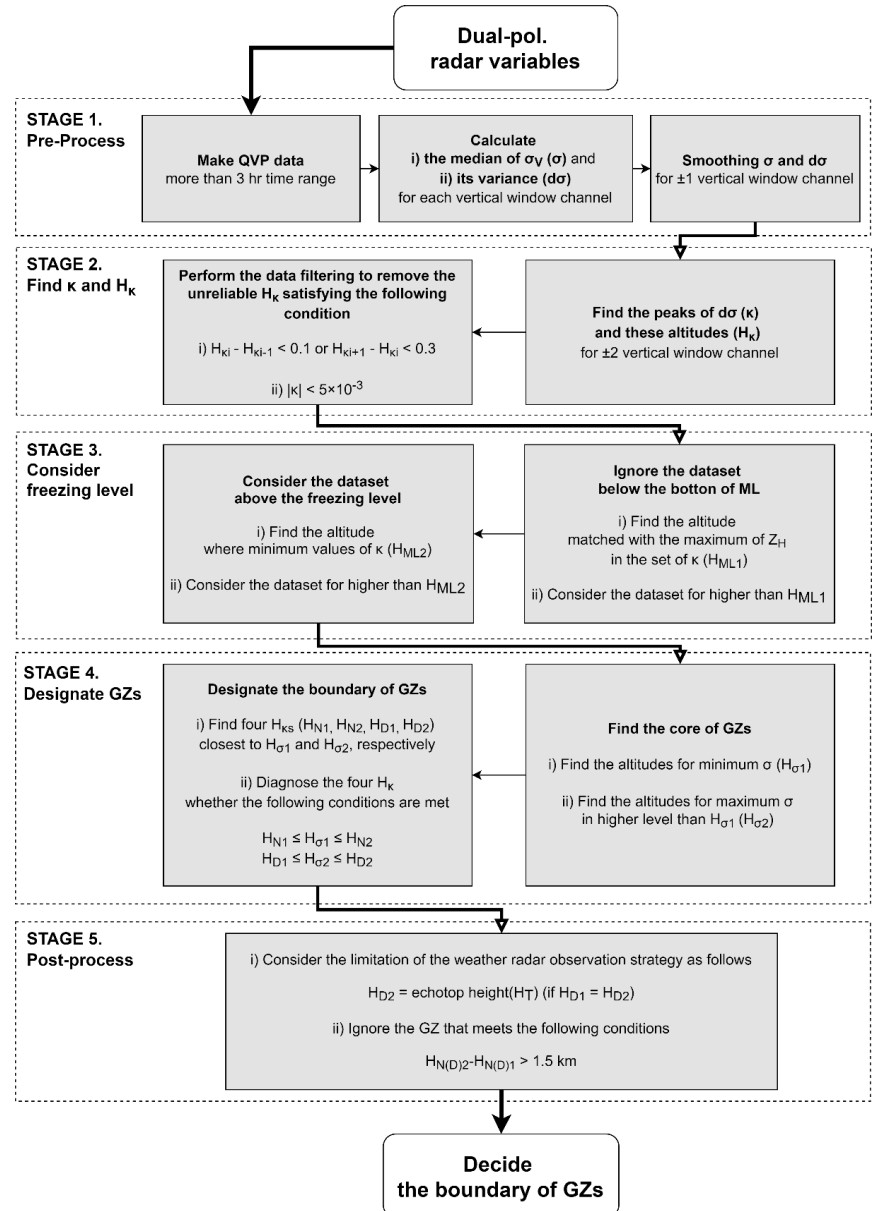


Figure 3. The algorithm of GZDA.




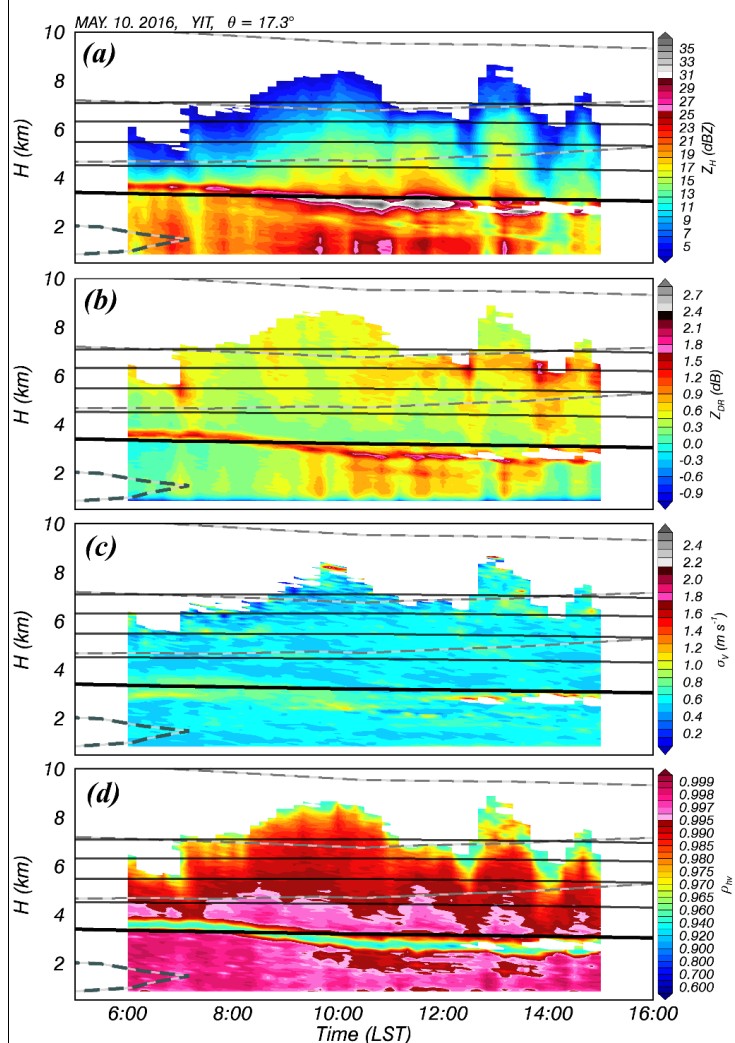


Figure 4. QVP of (a) $Z_H$, (b) $Z_{DR}$, (c) $\sigma_v$, and (d) $\rho_{hv}$ in the representative non-winter precipitation

(10[th] May 2016). Solid black and dashed blue curves in the background represent T and $v$, respectively,

obtained from the MSM reanalysis data. The temperature profile is expressed down to -20ºC.



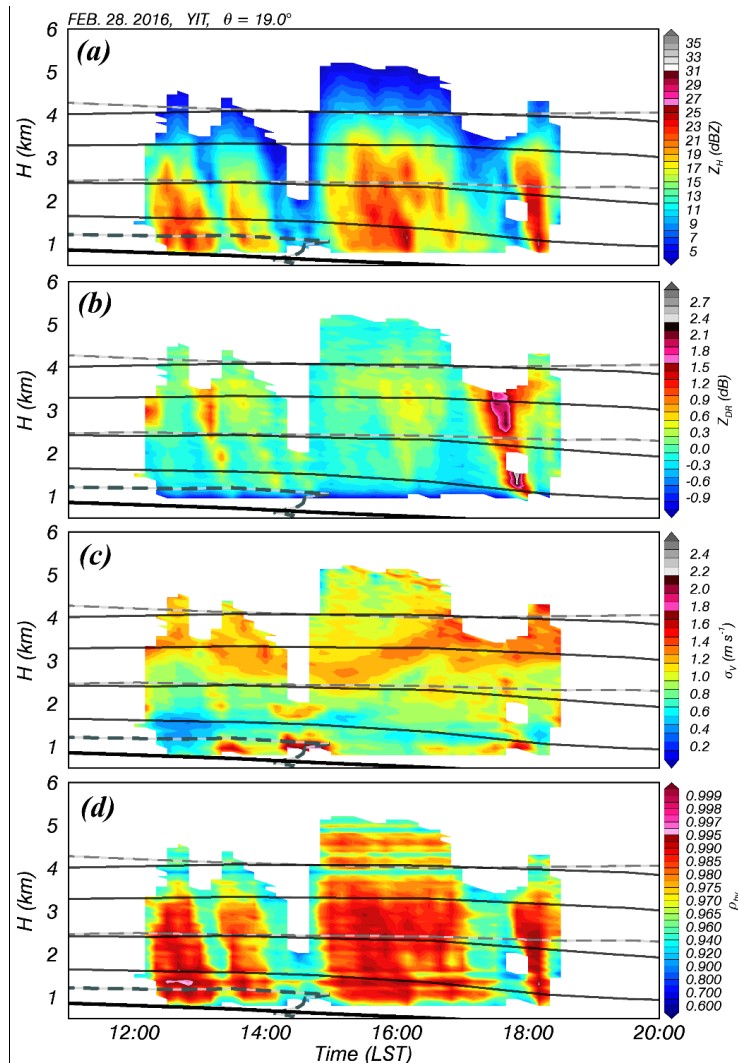


Figure 5. Symbols and colors are the same as in Figure 4 but for the representative winter

precipitation (28[th] Feb 2016).

.

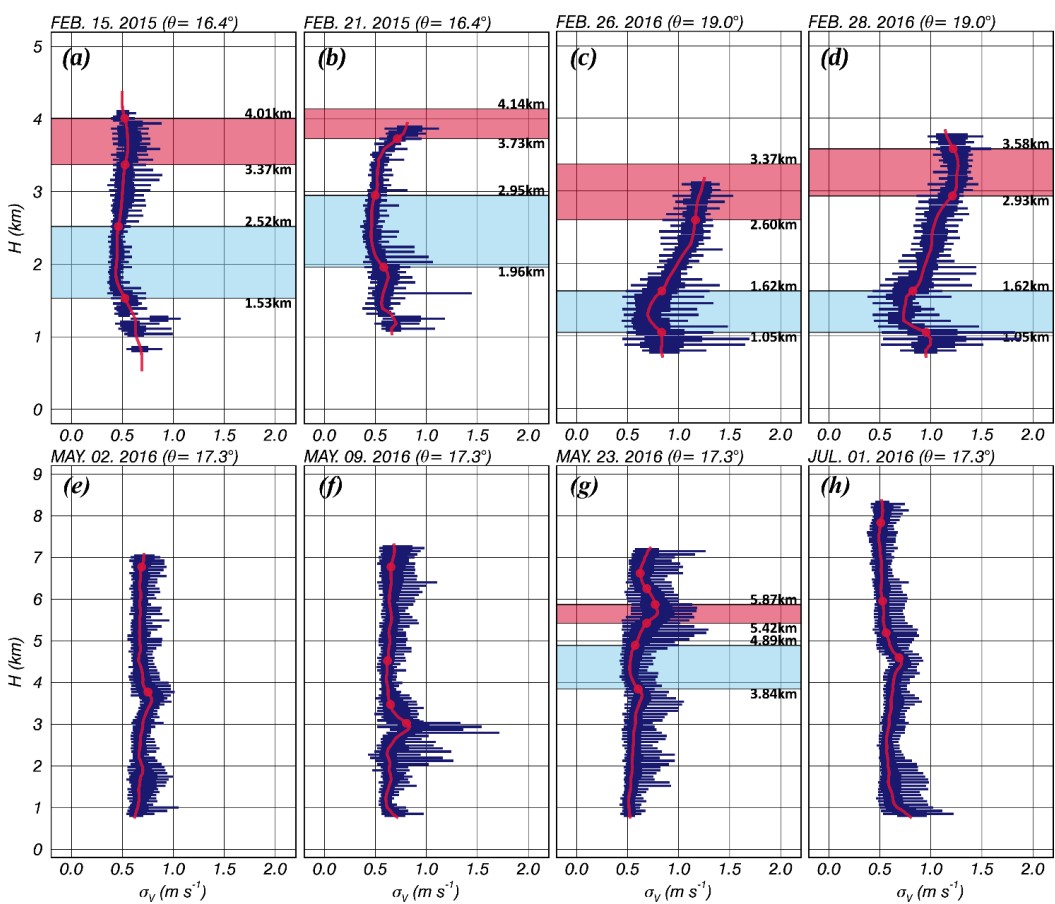

Figure 6. Vertical profiles of $\sigma_v$ quartiles for each height resolution level. The solid red lines indicate averaged $\sigma_v$ values, while red circles represent the peak point of d$\sigma_v$. Red and blue shaded areas are the areas of the DGZ and the NGZ, respectively, as determined by GZDA.






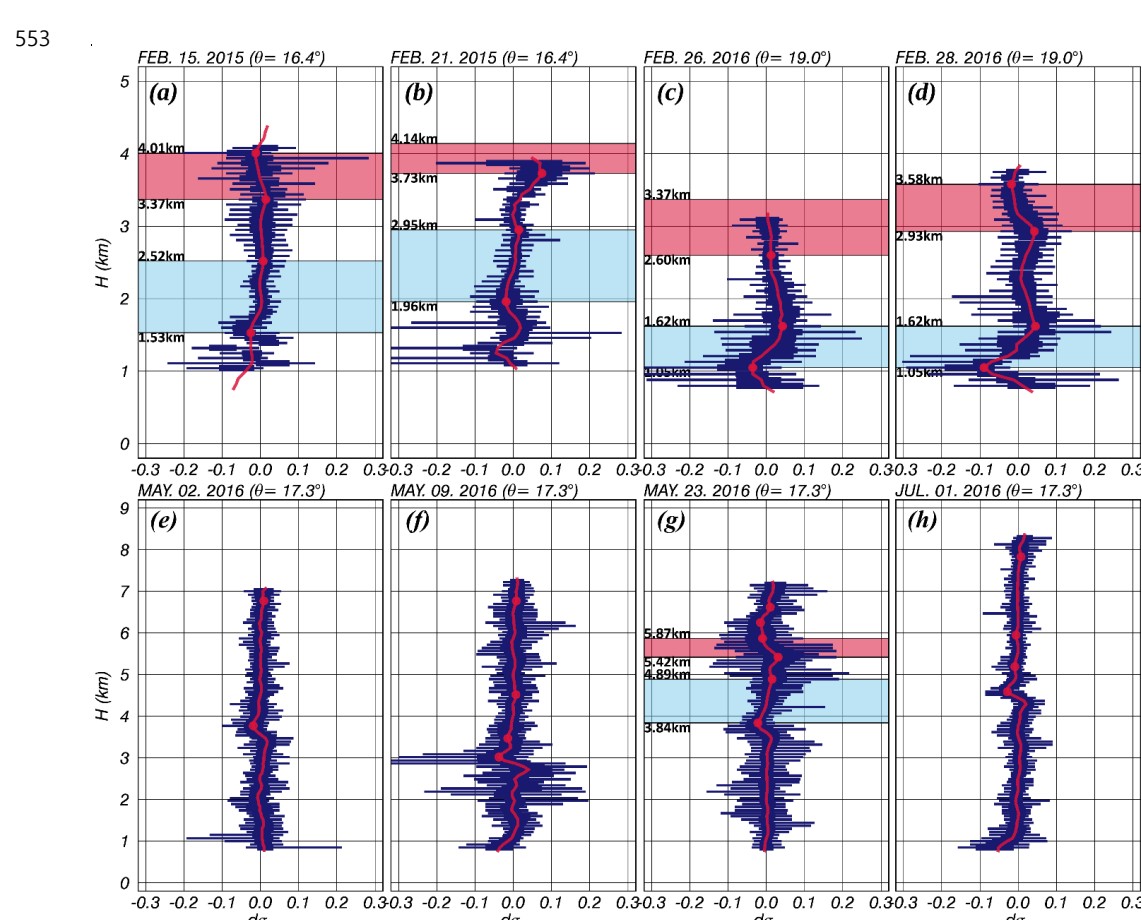


Figure 7. Symbols and colors are the same as in Figure 6 but for the dσ$_v$.




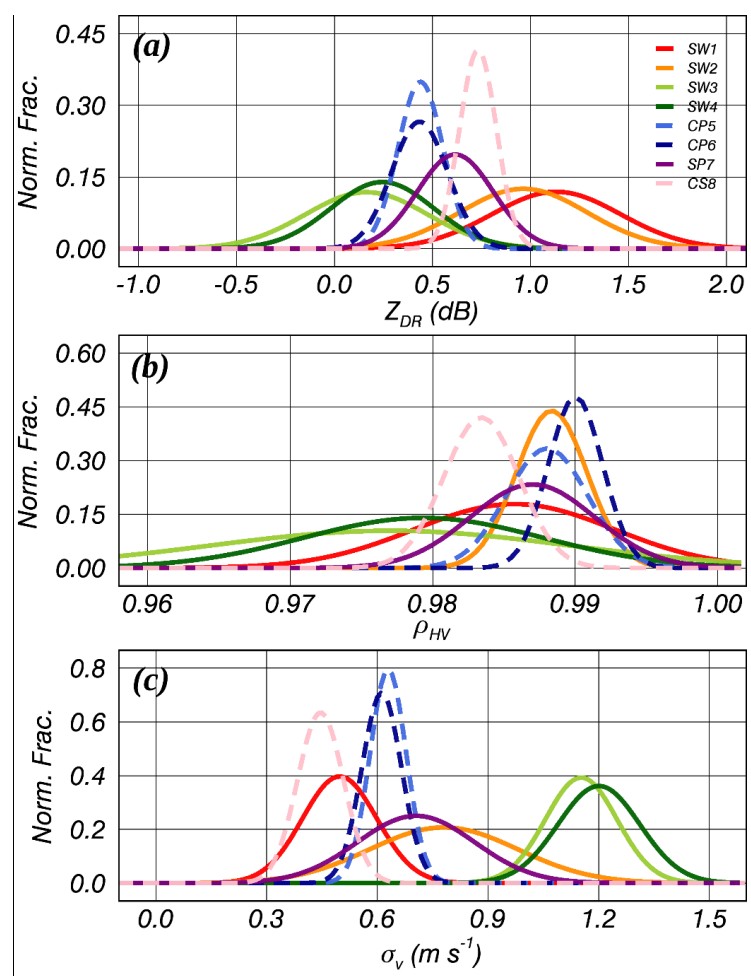


Figure 8. Normalized Gaussian distribution of (a) $Z_{DR}$, (b) $\rho_{hv}$, and (c) $\sigma_v$ in the DGZ for analyzed

cases. The solid and broken line represents stratiform and convective types, respectively.




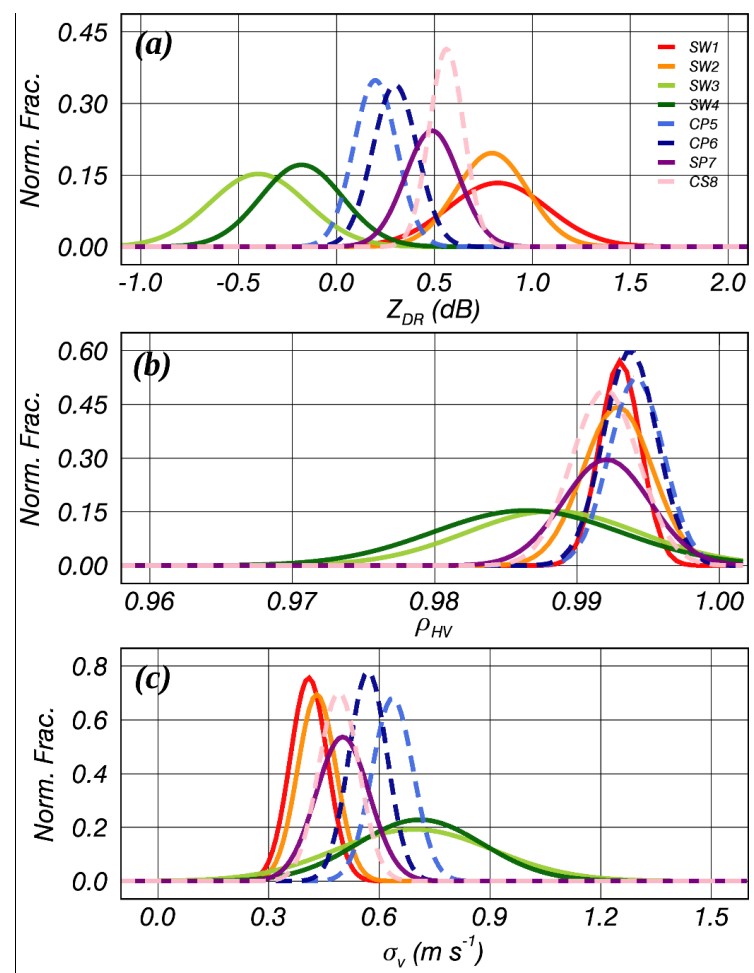

Figure 9. Legend is the same as in Figure 8 but for the NGZ.

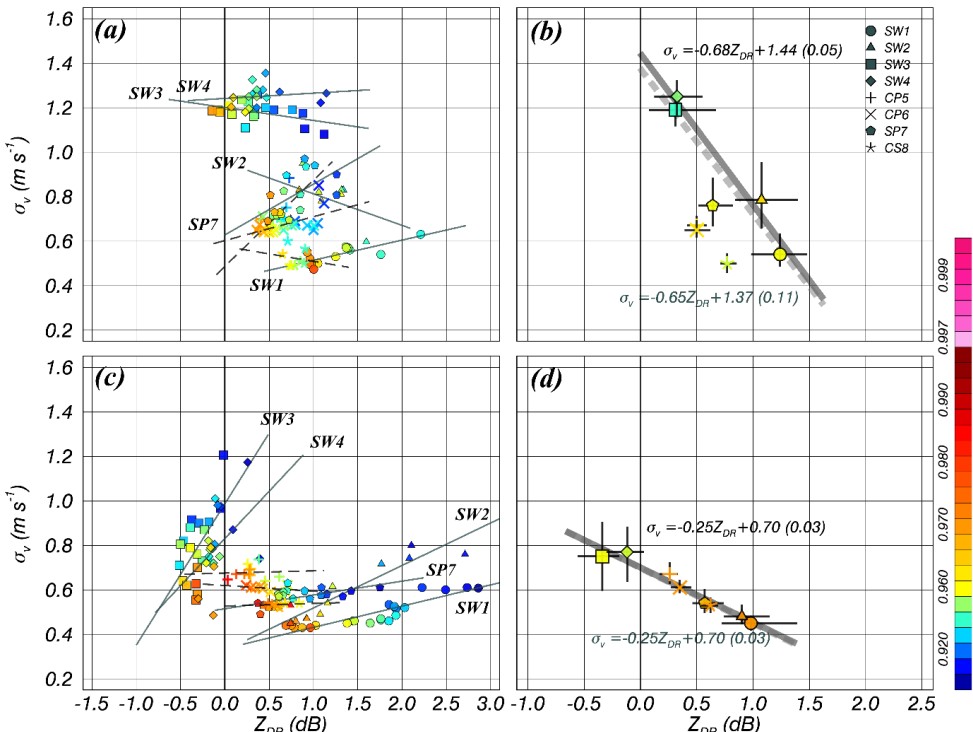

Figure 10. Scatter plot of the averaged $\sigma_v$ –$Z_{DR}$ by rank of $\rho_{hv}$ for the DGZ (a & b) and the NGZ (c & d). The $\sigma_v$ –$Z_{DR}$ were averaged (a & c) within each case and for each case (b & d), respectively. Solid lines represent a regression line, and the shape of crosses overlapped with each symbol in (b & d) indicate the 1$^{st}$ to 3$^{rd}$ quartiles for variables on each axis. Thick solid and broken lines represent a regression line for the winter and whole stratiform cases, respectively. The parentheses in relationships are the RMSE.



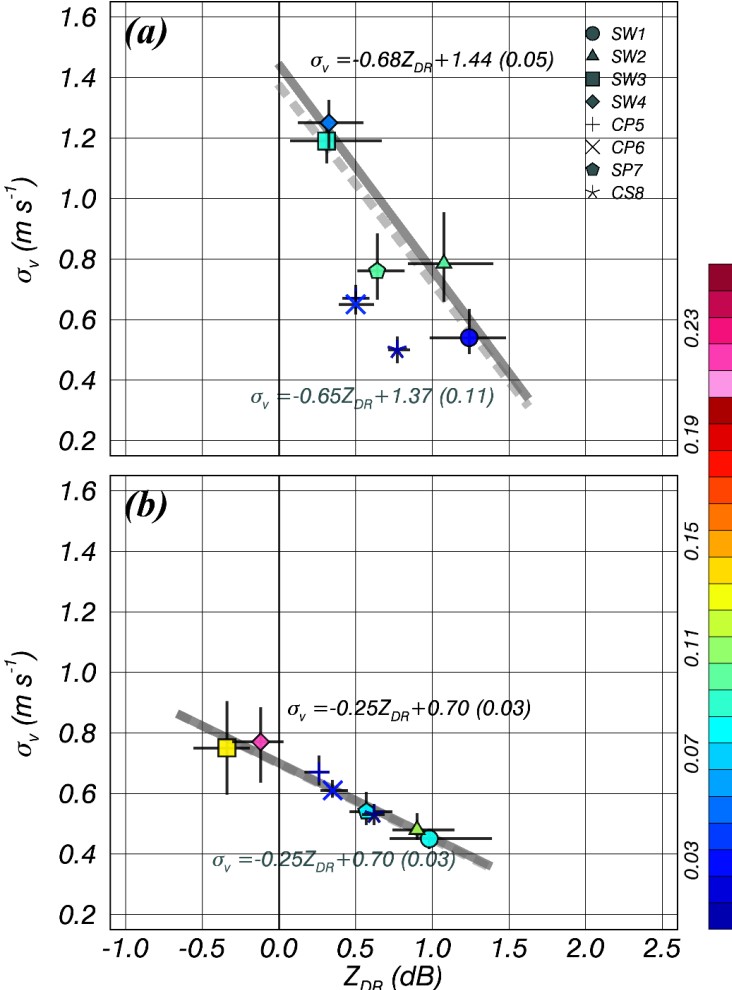

Figure 11. Scatter plot of the averaged $\sigma_v$ –$Z_{DR}$ for the case in (a) the DGZ and (b) the NGZ. The

colors on the symbols represent the $d\sigma_{MAX}$. Thick solid and broken lines represent a regression line for

the winter and whole stratiform cases, respectively. The shape of crosses overlapped with each symbol

indicates the 1st to 3rd quartiles for variables on each axis. The parentheses in relationships are the

RMSE.



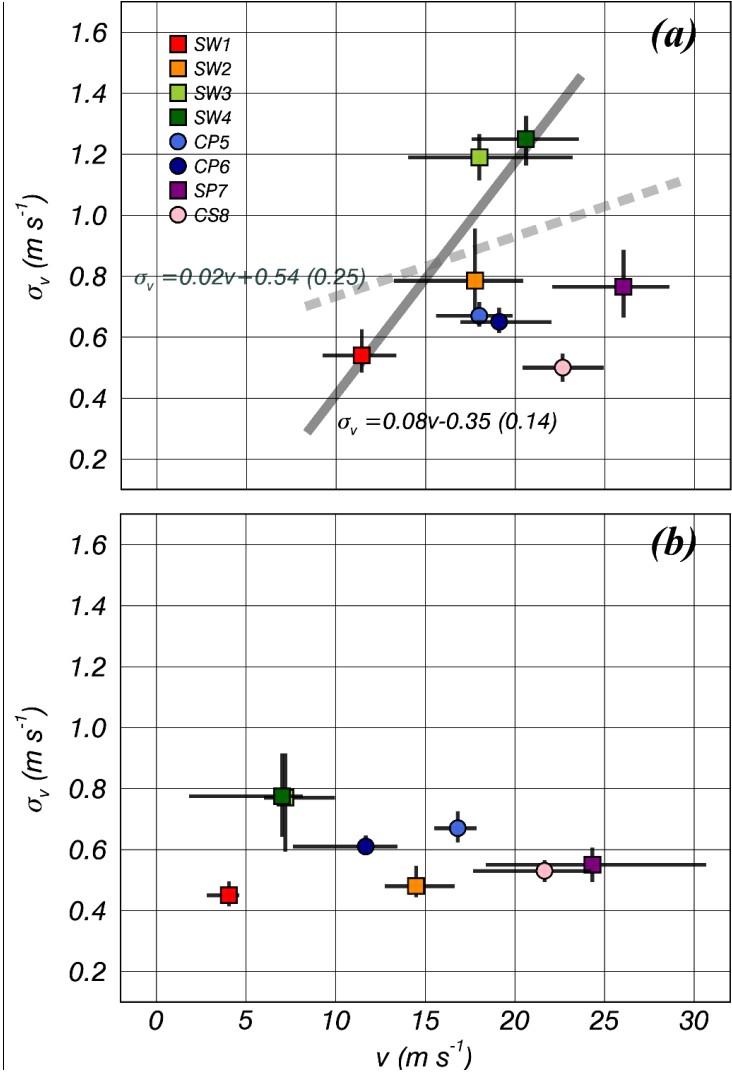

Figure 12. Scatter plot of the averaged $\sigma_v - v$ for the case in (a) the DGZ and (b) the NGZ. The

colors of the symbol represent the case. Thick solid and broken lines represent a regression line for the

winter and whole stratiform cases, respectively. The shape of crosses overlapped with each symbol

indicates the 1st to 3rd quartiles for variables on each axis. The parentheses in relationships are the

RMSE.



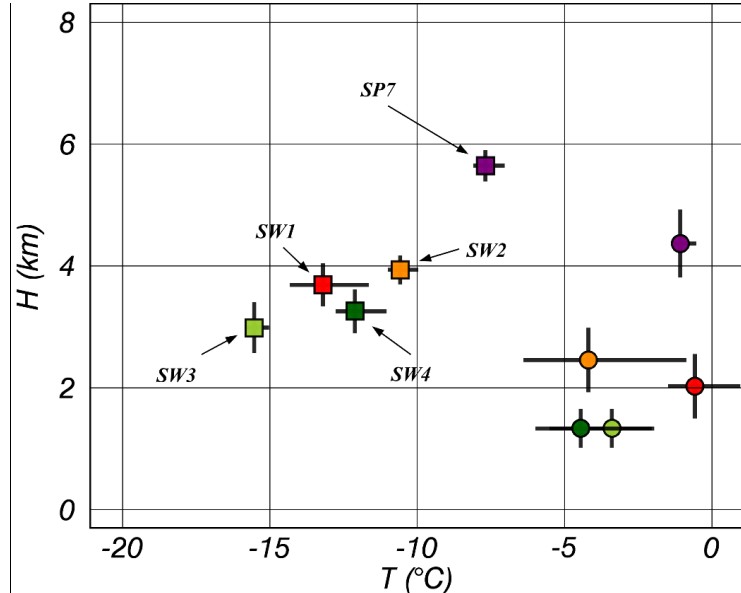

584

Figure 13. Scatter plot of the averaged H–T for the stratiform case in the DGZ (square symbols)

and the NGZ (circle symbols). The colors on the symbol represent the case. The shape of crosses

overlapped with each symbol indicates the 1st to 3rd quartiles for variables on each axis.

588





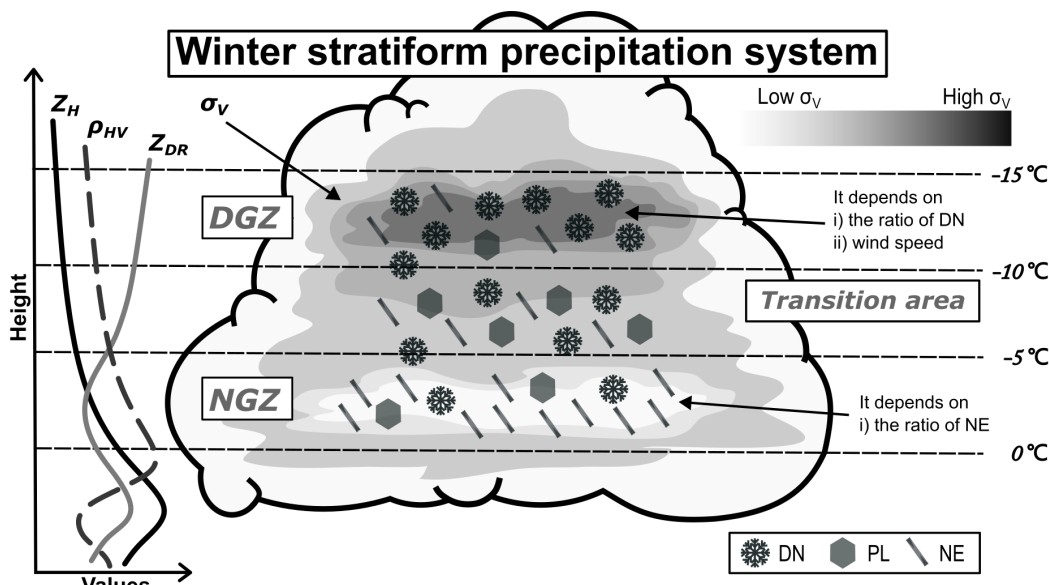

Figure 14. Schematic representation of the vertical structure of radar variables and the expected distribution of solid hydrometeors in a stratiform precipitation system above the melting layer.