# Peer review of "Statistical Analysis on the Estimations of Solid Hydrometeors Growth Zones and Their Weather Conditions Using Radar Spectrum Width"

_EGUsphere, 2023_

## Referee Comment (RC2)

Review of "Statistical Analysis on the Estimations of Solid Hydrometeors Growth Zones and Their Weather Conditions Using Radar Spectrum Width", by Suh and Coauthors, egusphere-2023-947.

This is an interesting article with the goal of providing real-time guidance using radar for aircraft safety awareness of potentially adverse weather conditions. The article is well-written overall. My impression-and comments, have to do with some weaknesses in the representation of the physics of ice particle growth and aggregation, and fallout. My comments appear below. I have more comments but these are the primary ones I'd like the authors to consider.

Abstract. $\sigma_v$ and DN relationship. Isn't it possible that the high value of $\sigma_v$ is due to there being a combination of dendrites and dendritic aggregates-dendrites fall at about 25 cm/s whereas aggregates fall at about 150 cm/s at altitude. And, in the NE region, the particles could all be aggregates and therefore have a low value of $\sigma_v$. I suggest you consider the following. Look at the Doppler velocities, adjusted for vertically falling, and see if the growth zones you derive are consistent with single crystals or aggregates, and whether the velocities are consistent with dendritic aggregates in the dendritic growth zone and needle aggregates in the needle growth zone.

Line 25. Needle type snowflakes. In general, I disagree with this. What happens when dendrites, formed at temperatures of -10 to -20C, and with a large cross-sectional area to aggregate, fall through the needle growth regime. The needles stick on the dendritic aggregates. Yes, needle aggregates can occur, but in the cases you show, dendrites from aloft will aggregate and fall through the needle growth layer.

204-206. Supercooled droplets are often found in wintertime storms-see below. I disagree with this sentence. It needs to be rewritten.

[Figure]

Figures 4, 5 and 6. A temperature scale is needed, maybe a second ordinate axis on the left side.

215-229. You mention what the SW numbers refer to earlier. You may want to mention it again here.

4. Discussion. 331-333. The physics behind this sentence is not correct. The growth rate is dependent upon the temperature, relative humidity, and difference in the water vapor density at this relative humidity and the vapor density at saturation with respect to ice. Water vapor density does not depend on the altitude but rather the temperature. An obvious example is where there is a temperature inversion or when the relative humidity is low near the surface. This section needs to be rewritten.

**Minor Comments**

Change "Freezing Level" to "Melting Level" everywhere

204: can only have. usually have solid hydrometeors.

247: Contrarily >Conversely

345: will? or is possible?

Andy Heymsfield, NCAR